# Recent Developments in Botulinum Neurotoxins Detection

**DOI:** 10.3390/microorganisms10051001

**Published:** 2022-05-10

**Authors:** Christine Rasetti-Escargueil, Michel R. Popoff

**Affiliations:** Institut Pasteur, Université de Paris, Unité Toxines Bactériennes, F-75015 Paris, France; popoff2m@gmail.com

**Keywords:** botulinum neurotoxins, detection, botulism, in vitro, in vivo, cell-based assays, countermeasures

## Abstract

Botulinum neurotoxins (BoNTs) are produced as protein complexes by bacteria of the genus *Clostridium* that are Gram-positive, anaerobic and spore forming (*Clostridium botulinum*, *C. butyricum*, *C. baratii* and *C. argentinense* spp.). BoNTs show a high immunological and genetic diversity. Therefore, fast, precise, and more reliable detection methods are still required to monitor outbreaks and ensure surveillance of botulism. The botulinum toxin field also comprises therapeutic uses, basic research studies and biodefense issues. This review presents currently available detection methods, and new methods offering the potential of enhanced precision and reproducibility. While the immunological methods offer a range of benefits, such as rapid analysis time, reproducibility and high sensitivity, their implementation is subject to the availability of suitable tools and reagents, such as specific antibodies. Currently, the mass spectrometry approach is the most sensitive in vitro method for a rapid detection of active or inactive forms of BoNTs. However, these methods require inter-laboratory validation before they can be more widely implemented in reference laboratories. In addition, these surrogate in vitro models also require full validation before they can be used as replacement bioassays of potency. Cell-based assays using neuronal cells in culture recapitulate all functional steps of toxin activity, but are still at various stages of development; they are not yet sufficiently robust, due to high batch-to-batch cell variability. Cell-based assays have a strong potential to replace the mouse bioassay (MBA) in terms of BoNT potency determination in pharmaceutical formulations; they can also help to identify suitable inhibitors while reducing the number of animals used. However, the development of safe countermeasures still requires the use of in vivo studies to complement in vitro immunological or cell-based approaches.

## 1. Introduction

Botulinum neurotoxins (BoNTs) are produced as protein complexes by bacteria of the genus *Clostridium* that are Gram-positive, anaerobic and spore forming (*Clostridium botulinum*, *C. butyricum*, *C. baratii and C. argentinense* spp.). BoNTs show a high immunological and genetic diversity. Indeed, BoNTs are divided into nine toxinotypes (A, B, C, D, E, F, G, H or F/A, X) based on their neutralization by specific corresponding antisera, and their actions on different substrates. In addition, each toxinotype is subdivided into subtypes based on amino acid variations. Currently, 41 subtypes have been identified [1]. While types A, B, E and F are mainly responsible for human botulism, toxinotypes C and D are associated with animal botulism mainly occurring in birds and cattle. The various BoNTs toxinotypes and subtypes interact with distinct membrane receptors, and cleave different intracellular SNARE proteins (soluble N-ethylmaleimide-sensitive factor attachment receptor), VAMP (vesicle associated membrane protein)/synaptobrevin, synaptosomal-associated protein 25 (SNAP-25), and syntaxin, at different cleavage sites [2,3]. The rapid progress in genomic information has revealed the presence of *bont* related sequences in non-clostridial strains, such as *bont*/Wo or *bont*/I detected in the genome of the *Weisenella oryzae*, a bacterium of fermented rice; the *bont*/J (*ebont*/F or *bont*/En) found in the genome of a *Enterococcus faecalis* strain isolated from cow feces; Cp1 from *Chryseobacterium piperi* from sediment [4,5,6,7]; and the paraclostridial mosquitocidal protein 1 (PMP1) in a *Paraclostridium bifermentans* strain [8].

Botulism occurs in three major syndromes: food-borne botulism due to consumption of preformed BoNT in contaminated food, wound botulism, and botulism by intestinal colonization (infant botulism and adult intestinal toxemia). *C. botulinum* can grow in various food matrices and produce toxin as a result of inappropriate storage temperature in association with anaerobic conditions. Quantities as low as 30 ng BoNT can cause botulism. Wound botulism also occurs within injured tissues where the spores can grow and produce toxin in the tissues [9]. *C. botulinum* spores may result in a toxico-infection by colonization of the intestinal tract and in situ BoNT production. Children under the age of 1 year can develop infant botulism, since they are more susceptible to intestinal colonization by *C. botulinum* [10,11,12]. In addition, inhalational botulism can result from aerosolization of BoNT in rare cases of laboratory botulism, and iatrogenic botulism can result from the injection of BoNT overdoses after therapeutic or cosmetic use [13,14,15,16].

Botulism clinical symptoms are characterized by patients exhibiting flaccid paralysis, beginning with symmetrical cranial nerve palsies, followed by descending, symmetric flaccid paralysis of voluntary muscles; this may progress to respiratory compromise and death. Moreover, botulism symptoms include an inhibition of secretions from glands innervated by cholinergic neurons (lachrymal, salivary, and sweat glands), in addition to paralysis of the intestinal and urinary tracts [13]. Treatment of botulism includes meticulous intensive care with mechanical ventilation, when necessary, and administration of antitoxin as soon as the diagnosis is confirmed. 

Fast, precise, and reliable detection methods are paramount to monitor outbreaks and ensure surveillance of botulism. The botulinum toxin field also encompasses therapeutic uses, which notably require precise determination of BoNT types, basic research studies, and biodefense issues. This review presents currently available detection methods, and new methods with the potential of offering enhanced precision and reproducibility.

## 2. Applications of BoNT Detection

### 2.1. Clinical Presentation of Botulism

Botulism cases are initially investigated based on the clinical presentation provided by the clinicians, and only confirmed after thorough laboratory investigation of the presence of BoNTs and/or clostridial organisms able to produce BoNTs. Symptoms occur within 6–36 h after toxin ingestion, and it is challenging for clinicians to differentiate symptoms of botulism from certain neurological conditions such as Gillain-Barré syndrome, or cerebrovascular accidents. Very often, botulism presentation consists of gastrointestinal early symptoms, such as constipation, abdominal pain, nausea and vomiting; however, these types of symptoms may also be caused by microorganisms not related to *Clostridium botulinum*, hence delaying the proper diagnosis based on flaccid paralysis [17]. 

Rapid administration of the antitoxin is currently the only efficient therapy to treat the disease, which implies a rapid and accurate diagnosis since the laboratory confirmation takes a few days. Medical treatment also includes supportive care, intubation, and mechanical ventilation when necessary. However, once the toxin has entered the neurons, intoxication remains irreversible, and no post-exposure therapy exists [18].

### 2.2. Laboratory Confirmation

The detection of BoNT, or identification of the producing organism, is a requirement for laboratory confirmation of botulism. However, the current gold standard assay, the mouse bioassay (MBA), is causing ethical concerns and is difficult to standardize. In order to address this challenge, many in vitro or cell-based assays have been developed to detect BoNTs or characterize the BoNT-producing organism using PCR, to reduce animal use or even replace the MBA. Nevertheless, many of the recent assays are not fully validated for the detection of all toxinotypes in clinical specimen or food matrices. BoNT detection is currently performed using the in vivo MBA routinely used to detect BoNTs in serum, feces of patients or in suspected food or environmental samples. BoNT detection in the serum of suspected patients is the most direct way to confirm a diagnosis of botulism. However, very sensitive methods of BoNT detection are required to identify the minute BoNT amounts in sera [9]. The MBA is also able to detect the wide spectrum of BoNT subtypes described in the literature, but it cannot distinguish the individual subtypes [16,18]. 

Many DNA-based methods have been developed to identify the presence of *C. botulinum* or spores in food, clinical or environmental samples, and study the phylogeny of outbreak strains to complete epidemiological investigations [10,17]. However, actual detection of preformed BoNT in food suspected to be responsible for a botulism outbreak is required to support the diagnosis of botulism.

### 2.3. Food Industry: Survey of Food Safety

Food safety regulations have been established by the World Trade Organization (WTO) to ensure the safety of food products worldwide and reduce risks to human health. Risk assessment in parallel with scientifically based food safety criteria have been established to assess the appropriate level of protection. The microbiological criteria are reviewed at regular intervals to account for new information on, for example, new scientific knowledge including newly identified pathogens and infectious doses. Several food safety regulations have been established around the globe to ensure safe food processing and decrease the incidence of food poisoning. Food safety has become a growing concern in Asia; major Asiatic countries have now established regulatory agencies to implement food safety regulations [19]. *C. botulinum* of Group I are widely spread across the environment, and pose the risk of food-borne botulism if they enter into the food chain, leading to consumption of pre-formed BoNT in contaminated food. The high thermally resistant spores produced by group I *C. botulinum* strains cause great concern in low-acid canned food. Thus, sensitive detection of BoNT in food samples represents the most reliable approach for source-tracking. Subsequent regulatory decisions, such as batch recalls or seizure, are based on laboratory methods confirming the presence of BoNT as a health hazard [17,18,19,20].

Detection of BoNT in food or water is also an important aspect in biodefense. BoNT is classified as one of the most potentially dangerous bioweapons in category A of the classification of the Centers for Disease Control and Prevention (CDC) of Atlanta (Classification of biological agents according to the Centers for Diseases Control and Prevention. http://www.bt.cdc.gov/agent/agentlistcategory.asp, accessed on 1 January 2022). BoNT could be spread through food, water or aerosols [21,22].

### 2.4. Pharmaceutical Industry Uses, Needs for Precise Quantification of Active BoNT

BoNT/A and BoNT/B are available as licensed pharmaceutical products to treat cervical dystonia, blepharospasm, spastic conditions, hyperhidrosis, and in an increasing number of other medical indications. BoNT is also used as a cosmetic to reduce facial lines. Safe dosing of BoNT products requires accurate and reliable measurement of potency before batch release onto the market. The reference test for potency testing is the MBA. Indeed, BoNT therapeutic units are mouse lethal doses (LD50). The biological activity of BoNT has to be determined accurately for safety and efficacy of the product. Current requirements of the European Pharmacopea for potency determination stipulate that every production batch of BoNT must be tested using the MBA [23]. In the USA, the Food and Drug Administration (FDA) requires the submission of data on BoNT potency (using in vivo and/or approved in vitro methods), in addition to the safety, purity, sterility and other parameters for these products. The regulations are specified in the Code of Federal Regulations 21.

## 3. BoNT Detection

A variety of detection methods are applied based on different principles. Functional methods, such as MBAs and hemidiaphragm assays, are based on the three steps of intoxication by the BoNTs (binding to cell surface receptor, translocation into neuron cytosol and cleavage of intracellular substrates). Immunological methods and mass spectrometry are based on the detection of BoNTs proteins, while endopeptidase assays are based on the measurement of cleavage products resulting from BoNT proteolytic activity. The various detection principles are described in Figure 1.

### 3.1. Physical Methods

Various physical methods have been described for the detection of BoNTs, but only a few methods present suitable attributes to validate their performance in food samples, patients’ sera and in environmental samples [24,25].

#### 3.1.1. Immunological Methods: Immunoassays (ELISA)

The immunological methods represent the most rapid, sensitive and reproducible way to detect and quantify the BoNT proteins in various sample types (Table 1) [26,27]. The detection by ELISA (enzyme linked immunosorbent) method is suitably sensitive; however, it detects the active and inactive forms of the toxin without distinction of the fully active BoNT. A collaborative study organized by the FDA and the CDC (Centers for Disease Control and Prevention) to assess the performance of ELISA kits to detect BoNT toxinotypes A, B, E, and F in various food matrices has established the suitability of ELISA kits to detect BoNTs in food samples [28,29]. The most recent and sophisticated version of the ELISA method, the Luminex assay, uses microsphere beads conjugated to antibodies; it has shown better limits of detection than the MBA [30,31].

#### 3.1.2. Mass Spectrometry Methods

The type and subtype identification on BoNT/A1 and A2 isolated from spiked milk were made possible using mass spectrometry (MS) analysis. This method allows the analysis of BoNT with toxin type identification within a few hours, and subtype identification within 24 h [39]. Detection of minute amounts of BoNT in human serum using MBA is the most direct way to confirm a diagnosis of botulism, but it was shown that the MS method can provide information on the presence of toxin within a matter of hours instead of days [40].

The matrix-assisted laser desorption ionization time-of-flight (MALDI-TOF) MS method combined with 16S rRNA sequence analysis can be implemented for species identification of *Clostridium* spp. Isolates, and is of public health importance [41]. A MALDI_TOF-based method has been proposed for simultaneous detection of BoNT/A, BoNT/B, ricin and staphylococcus enterotoxin B (SEB) [42]. A liquid chromatography-tandem mass spectrometry (LC-MS/MS) alternative approach was developed to detect BoNT/A in honey, using a reference labelled protein (limit of detection 9.4 ng/mL). This method can be useful in the monitoring of food safety, and more specifically to avoid infant botulism [43]. A LC-MS/MS method coupled to an immunocapture with antibodies against BoNT/A light chain (L) allows the detection and identification of each BoNT/A subtype (limit of detection 120–150 mouse lethal doses) [44]. 

The MS-based physical methods precisely detect BoNT types and subtypes. However, the sensitivity is lower than that of functional methods.

### 3.2. Functional Methods

#### 3.2.1. Biochemical Methods: Endopeptidase Assay

The measurement of the catalytic activity of the L domain of BoNTs represents a precise method to detect and identify BoNTs, since the substrate and cleavage sites are unique to each toxinotype. However, this type of method detects only the biochemical activity of the L chain.

##### Detection of Cleavage Products by Mass Spectrometry (MS-Endopeptidase Assay)

The mass spectrometry (MS) method can be exploited to identify BoNT cleavage products, given that each individual substrate sequence can be attributed to the action of one specific BoNT toxinotype. The excellent performance of these assays has now been well-established in complex food matrices and in clinical samples [45,46,47] (endopeptidase-MS assay or Endopep-MS). Kalb et al. have optimized the Endopep-MS format by addition of the immunoaffinity enrichment step, enabling the Endopep-MS assay to reach a similar or better limit of detection than the mouse bioassay [48]. This method was further optimized to decrease the limits of detection for spiked serum samples from 0.5 to 0.1 mouse LD50, and from 0.5 to 0.2 mouse LD50 for spiked stool [49]. 

Dr. Kalb’s team has further improved the sensitivity of the method by adding an extension of the N-terminus of the peptide substrate, in addition to selective substitutions of amino acids at the scission bond and at various other positions of the peptide sequence in order to improve its resistance to nonspecific cleavage in biological matrices, particularly in stool samples. A new peptide substrate for improved detection of BoNT/A was also developed, leading to a 10-fold increase in the assay’s sensitivity, both in buffer and in clinically relevant samples [50]. However, more research is still needed to detect and differentiate the many BoNT subtypes [51,52]. Furthermore, for improved detection of BoNT/G, peptide substrates were optimized by single and multiple substitution(s) and terminal modifications, resulting in a toxin detection level of 0.01 mouse LD50 [53].

To prevent false negative findings from occurring during botulism outbreaks, it is paramount to test the peptide substrates against all BoNT subtypes when the BoNT subtype is not known [48]. The Endopep-MS method has been successfully modified to analyze cattle, horse, and avian liver samples by introducing a salt washing step and a protease inhibitor cocktail. These modifications have resulted in improved sensitivities for BoNT-C and C/D, which are among the most prominent toxinotypes for animal botulism during the diagnosis from outbreaks in Sweden [54]. 

##### Detection of Cleavage Products by Antibodies against Neoepitopes: Endopep-ELISA

The enzymatic activity of BoNTs can also be measured using an immunoassay platform. This platform was first optimized as a highly specific, sensitive, robust and reproducible immunoassay suitable for routine monitoring of BoNT/A products by Dr. Sesardic’s team in 2008. The assay sensitivity was increased 50-fold using an optimal 0.5% Tween 20 concentration with 0.1% albumin, and by removing the pre-activation/reduction step. Detection of 0.01 LD50/mL BoNT/A was achieved using a linear dose response between 0.1 and 1 LD50/mL [55,56].

This technique has also shown the potential to detect three BoNT toxinotypes (A, B, and E) by measuring their enzymatic activity in an immunoassay [34]. The Endopep-ELISA uses monoclonal antibodies (mAbs) specifically targeting neo-epitope(s) generated during the cleavage of target substrates [57,58]. An innovative suspension array technology has been fully developed to detect all clinically relevant BoNT toxinotypes, A through F, based on a Luminex platform. This assay has been shown to be highly sensitive and applicable to clinical and food matrices. To this aim, a comprehensive panel of Neo-mAbs recognizing the individual cleavage products of the synaptic substrates, SNAP-25, or VAMP-2 after cleavage by BoNT/A to BoNT/F, was generated and characterized. This type of in vitro functional technique has the potential to fully replace the MBA, thanks to an excellent sensitivity and robust measurements in clinical matrices. The detection of BoNT/A to F with specific mAbs reached a sensitivity in the range of 0.3–80 pg/mL [59]. 

##### Detection of Cleavage Products by Antibodies against Neoepitopes Using Immunosensors and Förster Resonance Energy Transfer (FRET) Assays

An automated sensor assay was developed to measure the endopeptidase activity of BoNT/A. Recombinant SNAP-25 was coupled to the sensor surface of a surface plasmon resonance (SPR) system, and samples containing BoNT/A were injected over the substrate sensor. The substrate cleavage was monitored by measuring binding of an mAb (mAb10F12), recognizing specifically the SNAP-25 neoepitope exposed upon cleavage. This SNAP-25-chip sensor assay specifically detected BoNT/A at 1 LD50/mL in 5 min, and 0.01 LD50/mL in 5 h. This label-free method is 100 times more sensitive than the MBA, and can be used for rapid read-out of BoNT detection in environmental samples and quality control of pharmaceutical preparations [60]. For the detection of BoNT/B, a chip-containing synaptic vesicle membrane for the detection of VAMP cleavage was used to allow the detection of BoNT/B at a picomolar level [61]. Bead-based immunosensors have also been developed for the rapid and quantitative detection of biological toxins using nano-immunosensors, by coating SNAP-25 on fluorescent probe particles. Gold nanoparticles, (AuNP)-conjugated antibodies, attached to the probe particles when SNAP-25 proteins were cleaved by BoNT/A [62,63].

In addition to the immunosensors, the combination of the endopeptidase assay with the FRET assay that utilizes fluorescence donor and fluorescence acceptor (or quencher) offers a very powerful and sensitive assay to detect BoNT activity with simplicity, rapidity, cost effectiveness and readiness for scale up. The main advantage of the FRET assay is its ability to detect enzymatically active toxins, unlike ELISA. The assay can be used to detect BoNTA in serum samples, as well as for pharmacokinetic and pharmacodynamic studies and inhibitor screening [64,65].

The ALISSA assay (assay with a large immunosorbent area) combines immunocapture with a fluorescent BoNT substrate. The monitoring of cleavage product by fluorescence uses conversion of fluorogenic peptide substrates to measure the intrinsic endopeptidase activity of bead-captured BoNT, allowing a very sensitive BoNT/A detection (10^−6^ pM) [66]. 

Another very innovative in vitro approach developed combines a nanosensor based on the use of nerve cell-mimicking nanoreactors (NMN) included into the microfluidic technology. This nanosensor recapitulates the three key BoNTs functionalities, and its integration into a microfluidic device allows the detection and quantification of BoNT in a much shorter time than the MBA (Figure 2) [67].

### 3.3. In Vivo and Ex Vivo Methods

The MBA has been the gold standard to detect and measure BoNT activity for decades, but the MBA is causing strong ethical concerns combined with data variability that have led to the need to reduce, refine, and replace this assay [68]. Use of the MBA can be reduced, thanks to new in vitro methods available; however, the MBA remains the reference tool for characterization of BoNT types known today, in addition to novel BoNTs [69]. The MBA can detect all BoNT types and subtypes, and is highly sensitive.

An alternative model for assessing muscle paralysis potency of BoNT therapeutic preparations has been developed in mice and rats by BoNT injection into the hind limb muscle. Paralysis of the digits is monitored, and the result is expressed as a digit abduction score [70,71]. However, the determination of digit paralysis is somewhat subjective, and this method is not easily quantifiable.

The mouse phrenic nerve assay is an example of an ex vivo model for testing BoNTs, having excellent agreement with the MBA for precise assessment of botulinum antitoxin neutralizing activity. This assay closely mimics in vivo respiratory paralysis, offering potential refinement. However, this technique is available in a limited number of laboratories, and is difficult to transfer to non-experienced laboratories [72].

### 3.4. Cell-Based Assays: Detection of Cleavage Products and Neurotransmitter Release Inhibition

Various alternative functional assays have been explored to reduce or replace MBAs, including cell-based assays to detect and measure the biological activity of BoNTs [73,74,75,76]. The cell-based assays involving neuronal networks can reach a similar sensitivity to the MBA, and also recapitulate the key steps of biological activity of BoNTs: binding, endocytosis, translocation and cleavage of SNARE substrate proteins. Additionally, cell-based assays can be used to assess neutralizing antibodies and offer unlimited cell sources using assay up-scaling. Nevertheless, the establishment of cell-based assays present many technical challenges, such as maintaining well-differentiated mouse or human neuronal cells derived from embryonic or induced pluripotent stem cells, and guaranteeing suitable reproducibility of assay sensitivity. Additionally, differentiated neuronal cells are directly impacted by the sample matrices, making this type of assay not directly applicable to challenging sample matrices. Nevertheless, neuronal cell-based assays have been approved by the FDA for potency testing of BoNT-based drug products [77], and for testing toxoids and antitoxin preparations [75,76,77,78]. 

The detection of SNARE cleavage products is one of the major endpoints used to detect BoNT activity within neuronal cell populations. This endpoint can be measured in the cell lysates using Western blot analysis, ELISA techniques or immunofluorescence [75,76,77,78,79]. 

Another important endpoint consists in measuring the inhibition of neurotransmitter release after exposure to BoNT. The neuronal cells are pre-loaded with radiolabeled neurotransmitter, followed by measurement of the released radioactivity at the basal state, and upon depolarization by KCl [80,81]. The use of a fluorescent dye as a marker of neurotransmitter release can also be used as an alternative [82,83]. As an example, mouse primary spinal cord cells loaded with ^3^[H]glycine allowed detection of BoNT/A at 0.01 pM [74]. In the differentiation of human embryonic carcinoma cells into mature neurons, analysis of inhibition of FM1-43 exocytosis by fluorescence imaging allows detection of BoNT/A at 10 ng/mL [83]. For SH-SY5Y cells differentiating into mature neurons, the sensitivity to BoNT/A reaches picomolar levels after a suitable differentiation procedure [82].

The cell lines are easy to maintain, but their sensitivity to BoNTs is affected by their cancerous origin. The early studies were performed on neuro-2a (mouse neuroblastoma) and PC12 (rat pheochromocytoma) cells; however, the cells’ sensitivity was in the nanomolar range, and the incubation times were quite long (48 to 72 h). The human neuroblastoma BE(2) M17 cell line presented also a sensitivity to BoNT/A in the nanomolar range by measuring the cleavage product using Western blot technique or the inhibition of neurotransmitter release [84]. The neuroblastoma cell line SH-SY5Y has been extensively studied for the development of more sensitive cell-based assays [85]. The differentiation of the SH-SY5Y cells using retinoic acid for 5 days followed by culture in medium containing BDNF and pre-incubation with GT1b, significantly improves their sensitivity to BoNT/A using the neurotransmitter release read-out. The sensitivity reached the low picomolar range [82]. 

In NG108 cells grown into optimized culture medium with retinoic acid, purmorphamine, TGFβ1 and GT1b, the sensitivity with BoNT/A monitored by detection of SNAP-25 cleavage by Western blot, and immune detection reached 7.9 pM [79,86].

One cell-based assay using the human neuroblastoma SiMa cell line has been developed by Allergan laboratories for the measurement of product potency. This assay is based on the sensitive detection of SNARE cleavage by BoNT/A, and has a sensitivity comparable to the MBA (in the picomolar range) [77]. Another group was able to develop an alternative cell-based assay for the characterization of BoNT/E antitoxins using the SiMA cell model. This assay measured the residual cellular activity of BoNT/E by a specific quantitative sandwich ELISA for its cleaved cellular target protein SNAP-25 [87]. This assay was applied in the assessment of neutralizing anti-BoNT/A antibodies [88].

The neuronal primary cells have been used since early times to elucidate BoNTs’ mechanism of action. Many primary cell types have been explored, such as cortical neurons, spinal cord neurons, hippocampal neurons or dorsal root ganglion cells [89,90,91,92,93,94]. Nevertheless, primary cells consist of a mixture of neurons and glial cells, thus they cannot be used as a pure neuronal population. After many assay developments, methods involving primary cells have now reached sensitivities to BoNTs that are comparable to the mouse bioassay’s sensitivity.

Stem cells can be differentiated into neuronal subpopulations, but more studies are still needed to reach pure neuronal populations and decrease batch-to-batch variability. Human neurons derived from human-induced pluripotent stem cells (hiPSC) provide a highly sensitive platform for BoNT potency determination, reaching a sensitivity close to 0.3 LD50 when using detection of SNAP-25 cleavage by Western blot analysis [73,95,96].

Mouse embryonic stem cell-derived motoneurons were shown to be as sensitive as mouse primary spinal motoneurons (picomolar range at 10 pM) when using the same principle of detection of SNAP-25 cleavage by Western blot analysis [75,97].

More recently, human motor neurons (MNs) were generated from induced pluripotent stem cells (iPSCs) and compared to the neuroblastoma cell line SiMa. In comparison with the mouse bioassay, human MNs exhibit a superior sensitivity to BoNT toxinotypes A1 (0.003–0.5 pM) and B1. However, SiMa cells exhibited much lower sensitivity (5.5 pM) than human MNs, and appeared unsuitable to detect any BoNT/B1 activity [98,99]. 

Further analysis of BoNT effects on functionality of endplate motoneurons was performed using optical stimulation combined with calcium imaging. In this study, the hiPSC-derived motoneurons co-cultured with skeletal muscle cells (motoneuron endplates) reached a sensitivity close to 0.6–0.5 pM by detection of SNAP-25 cleavage [100].

### 3.5. Cell-Based Assay: Measurement of Electrical Conductance Using Multi-Electrodes Arrays

The measurement of electrical conductance of a neuronal network with cultured neurons can be used to detect BoNT inhibitory actions. Live neural cultures can be differentiated directly onto microelectrode arrays (MEAs), allowing extracellular recording of the action potentials of neural networks. After 21 days in culture, embryonic stem cells form neuronal networks showing spontaneous bursting activity. Treatment with BoNT/A for 24 h results in a significant reduction in both spontaneous network bursts and average spike rates [101]. The activity of neuronal networks measured with MEAs can provide the basis for a valuable assay format for BoNT detection [102,103]. The measurement of synaptic activity by whole patch-clamp electrophysiology was performed in murine embryonic stem cell-derived neurons, which allowed for the detection of BoNT/A at a sensitivity of 0.005 pM [104].

### 3.6. Cell-Based Assay and Bioluminescence

A more sensitive SiMA cell line expressing luciferase NanoLuc-VAMP was developed for BoNT/B detection, using measurement of cleaved VAMP using luminescence measurement. This modified cell line reached a sensitivity close to 1 MLD [103]. In this assay, SiMA cells with reporter luciferase reached a sensitivity close to 100 pM for BoNT/A, and a lower sensitivity for BoNT/B [105,106].

The CANARY assay (Cellular Analysis and Notification of Antigen Risks and Yields) is based on B-cells expressing a membrane bound BoNT/A antibodies and aequorin, a calcium-sensitive bioluminescent protein. An immunocapture step uses beads coated with BoNT/A antibodies in order to trap the toxin from the sample. Then, BoNT/A induces antibody clustering on B-cell surfaces, which results in entry of Ca^++^ and activation of aequorin. Luminescence is measured with a luminometer. The sensitivities of this assay are 10 ng/mL with BoNT/A in buffer; 7.4 to 62.5 ng/mL according to various food matrices (milk, vegetables, meat, fish); and 171 ng/mL in egg matrix [107].

## 4. Discussion: Suitability of Each Method for Their Applications

To summarize the findings of this study, many different methods have been developed for the detection of botulinum neurotoxins. They all have potential to replace the MBA to address the three Rs principles: replacement, refinement and reduction in animal research and use [108]. Many in vitro methods show sensitivity levels lower than that of the MBA, as shown in Table 2 below; however, not all in vitro methods are able to distinguish the protein toxin from fully active BoNT.

The immunological methods offer a range of benefits, including rapid analysis time, reproducibility and high sensitivity. However, their implementation is subject to the availability of suitable tools and reagents, such as antibodies. In addition, immunological methods only detect the protein toxin itself, with no indication of its true functionality. The mass spectrometry approach is the most sensitive in vitro method currently available for a rapid detection of active or inactive forms of BoNTs. In addition, ELISA methods are easily transferable, and offer the potential for high-throughput analysis.

Note that these methods require inter-laboratory validation in different food, environmental, and clinical sample matrices before they can be implemented more widely in reference laboratories. 

As opposed to ELISA or MS, the endopeptidase ELISA or MS-based methods offer the rapid and sensitive detection of the cleavage activity. Nevertheless, they still require the use of a range of cleavage site-specific antibodies, which is a limitation considering the wide diversity of the BoNTs; in contrast, the fully functional assays do not require a range of specific antibodies. In addition, these surrogate in vitro models also require full validation before they can be used as a replacement bioassay of potency.

Cell-based assays using neuronal cells in culture recapitulate all functional domains of the toxin but they are still at various stages of development. Moreover, the cell-based assays that use primary cells are not sufficiently robust as a result of high batch-to-batch variability of their cultures. The models that employ differentiated mouse and human stem cells offer high sensitivity, and may provide the most useful alternative models for toxin assessment as well as for functional assessment of antibodies. The mouse phenic nerve assay is in excellent agreement with the mouse bioassay for precise assessment of botulinum activity and antitoxin neutralization; however, this method is hardly transferable to routine laboratories [72].

However, it is worth noting that cell culture assays are subject to interference and non-specific toxic effects, as a result of the presence of adjuvants and other ingredients in the matrices. Therefore, the in vivo model currently remains as the method of choice for detecting BoNT in parallel with immune-biochemical assays that correlate well with in vivo potency [56].

Nevertheless, significant progress has also been achieved by establishing functional muscle-nerve co-culture systems developed using hiPSC-derived motor neurons (hMNs) and human immortalized skeletal muscle cells. The recent implementation of optogenetics, combined with live calcium imaging, allows the monitoring of BoNTs’ intoxication on synaptic transmission in human motor endplate models [100]. These types of seminal studies demonstrate the potential to replace the mouse bioassay with well-designed, innovative, cell-based systems. Cell-based assays have a strong potential to replace the MBA for BoNT potency determination in pharmaceutical formulations; they can also help identify suitable inhibitors while reducing the number of animals used. The rapid progress in cell differentiation procedures as well in innovations of detection methods will support the replacement of the in vivo techniques; however, recently, the development of suitable countermeasures will still require in vivo studies in complement with the in vitro or cell-based approaches [69].

Recent developments of cell-based assays provide detection methods that are more sensitive than the MBA, and reflect all BoNT steps of activity; however, they have to be adapted for each BoNT type, whereas the MBA is sensitive to all BoNT types [107,109].

## Figures and Tables

**Figure 1 microorganisms-10-01001-f001:**
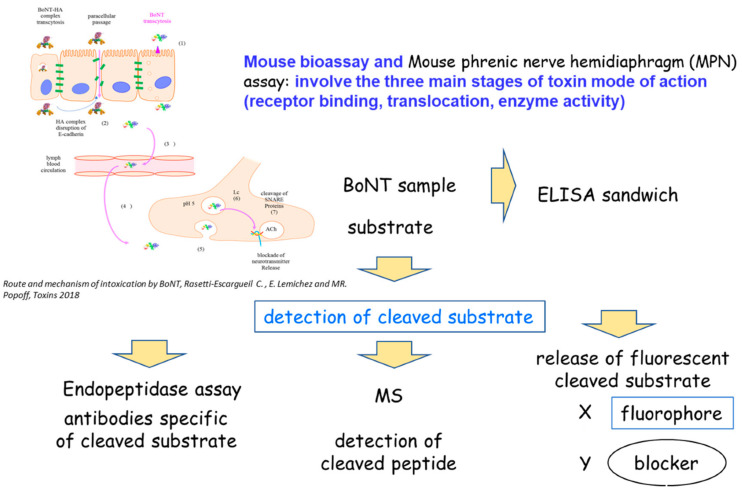
Detection principles.

**Figure 2 microorganisms-10-01001-f002:**
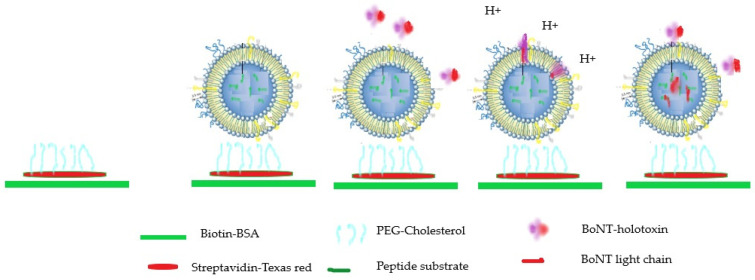
Sequential steps of microfluidic liposome immobilization and toxin detection. Surface modifications allow immobilization of NMN on the surface of the microfluidic channel. Botulinum neurotoxin is drawn through the channel and binds to the receptors on the liposomal membrane. Addition of the low pH buffer to the channel induces conformational changes of the receptor-bound toxin molecules and subsequent HC-mediated translocation of the LC into the liposome lumen. Inside the liposome lumen, LC proteolytically cleaves FRET peptide reporter molecules, which results in unquenching of the FRET pair and allows for fluorescent readout. The resulting increase in fluorescence intensity is directly influenced by the amount of physiologically active toxin molecules present in the original test solution, and therefore provides an estimate of its potency [67].

**Table 1 microorganisms-10-01001-t001:** Representative immunological-based methods of BoNT detection and their sensitivities.

BoNT Type	Method	Sample	Sensitivity	Reference
**A**	microfluidic double sandwich immunoassay	clinical serum samples	3 pg/mL (8 MLD/mL)	[32]
**A, B, E, F**	electrochemiluminescence with biotinylated antibodies bound to streptavidin-coated beads	food serum, urine,	50–100 pg/mL A, E 50 pg/mL B 100 pg/mL F 400 pg/mL	[33]
**A, B, E**	sandwich ELISA lateral flow immunoassay	buffer	A 10 pg/mL B 12 pg/mL E 107 pg/mL	[34]
**A**	flow cytometry with yeast displaying increased affinity scFv	buffer	15 pg/mL	[35]
**A**	ELISA	food sample	156–165 ng/mL	[36]
**A**	lateral flow immunoassay	buffer	10–50 ng/mL	[37]
**B**	sandwich ELISA	buffermilk	5 pg/mL 39 pg/mL	[38]
**A, B**	electrochemiluminescent assay	buffer	A 3–12 pg/mL B 13–17 pg/mL	[30]

MLD: mouse lethal dose.

**Table 2 microorganisms-10-01001-t002:** Methods of BoNT detection, sensitivities and durations.

Method Principles	Analysis Time	BoNT Toxinotype	Sensitivity	Benefits/Limitations	References
**Immunological methods: sandwich ELISA, electro-chemiluminescent assay**	6–7 h	A–F	2–176 pg/mL	Rapid detection/detection of active and inactive BoNTs, detection hampered by neurotoxin associated proteins	[25,26,27,28,29,30,31]
**Immunological methods: lateral flow assay,**	30 min	A–B	10–50 ng/mL (10,000–50,000 pg/mL)	Rapid detection/detection of active and inactive BoNTs, detection hampered by neurotoxin associated proteins	[25,34,37]
**Mass spectrometry**	5–8 h	A–F	0.1–1 pg/mL pg/mL	Rapid detection/detection of active and inactive BoNTs	[39,40,41,42,43,44]
**Endopeptidase ELISA based or MS based**	7–8 h	A–G	0.1–1000 pg/mL	Rapid detection/detection of cleavage only	[44,45,46,47,48,49,50,51,52,53,54,55,56]
**Immunosensors and FRET assays**	2–5 h	A	0.1–20 pg/mL	Rapid detection/detection of active and inactive BoNTs	[60,61,62,63,64,65]
**In vivo mouse bioassay**	4 days	A–F	1–10 pg/mL	Sensitive method detecting functional toxin but ethical concern, variability and duration	[23,68,69]
**Ex vivo methods** **hemidiaphragm test**	9–5 h	A–F	1–10 pg/mL	Sensitive method detecting functional toxin but ethical concern and technically demanding	[70,71,72]
**Cell-based assays** **human neurons from induced pluripotent cells and monitoring of SNAP-25 cleavage by Western blot**	3–5 days	A–E	0.003 pM–10 pM (0.55–1500 pg/mL)	Sensitive method detecting functional toxin but technically demanding	[73,74,75,76,80,81,89,90,95,96,97,98]
**Cell-based assays using differentiated cell lines**	3–5 days	A–E	5.5 pM–10 nM (825–150,000 pg/mL)	Sensitive method detecting functional toxin but technically demanding	[77,82,83,84,85,86,87,88,94,99,105,106]
**Electrical conductance assays**	1–3 days	A	25,000 pg/mL	Method detecting functional toxin but long and technically demanding	[86,87]

## Data Availability

Not applicable.

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
