# Peer review of "Recent Developments in Botulinum Neurotoxins Detection"

_microorganisms, 2022, doi:10.3390/microorganisms10051001_

Round 1

Reviewer 1 Report

This is a good paper and will create interest in your readers. this is especially important when clinicians want to know the precise amount of toxin in each vial rather than units. It gives good summary of basic science and molecular biology to the clinicians who is dealing with the botulinum toxin

Author Response

Dear Reviewer,

Thank you for the positive comments and for the consideration of our manuscript.

Sincerely yours

Dr C. Rasetti-Escargueil

Reviewer 2 Report

General comments:

This review provides a comprehensive overview of current BoNT detection methods. It could be improved by addressing the following comments.

Specific comments:

Abstract: Although this review covers several test methods, the abstract focus mainly on cell-based methods. I suggest revising the abstract to reflect the scope of the review.

Throughout the manuscript: Please define all abbreviations the first time they are used.

Page 2, line 70: Please replace “Diagnostic of” with “Diagnosis of”

Page 2, lines 80-83: Please mention that treatment also includes supportive care, intubation, and mechanical ventilation when necessary.

Page 2, lines 86-87: The sentence “In order to replace animal use…” is not clear. Maybe it is missing a few words? Please clarify.

Page 2, lines 90-92: The sentence repeats “BoNT detection in clinical specimens and foods”; please reword.

Page 2, line 84: Suggest changing the title of this section to “laboratory confirmation of botulism” and combining it with the previous section “Diagnosis of botulism”. Both sections refer to the same.

Page 3, lines 97-104: This section does not seem to fit here, as the review is about detection of BoNT, and this section refers to DNA based methods to identify C. botulinum.

Page 3, lines 130-138: the manuscript mentions the requirements of the European Pharmacopea for potency determination. What are the requirements outside the EU? How do pharmaceutical companies determine potency of BoNT in the US, for example?

Page 7, lines 262-20: Have these methods been used to test clinical specimens and/or foods? Please clarify.

Page 8, lines 288-297: Have these methods been used to test clinical specimens and/or foods? Please clarify.

Page 9, lines 374-384: Has this method been used to test clinical specimens and/or foods? Please clarify.

Page 10, lines 386-390: Has this method been used to test clinical specimens and/or foods? Please clarify.

Page 10, line 399: Please reword the title of this section.

Page 12, lines 409-436: Although the manuscript covers several test methods, the Discussion section focus mainly on cell-based methods. I suggest revising this section to reflect the more general scope of the review.

Author Response

Dear Reviewer,

Thank you for the revision of our manuscript, the comments are very helpful to improve the review.

Please find the replies to the specific comments:

General comments:

This review provides a comprehensive overview of current BoNT detection methods. It could be improved by addressing the following comments.

 Specific comments:

Abstract: Although this review covers several test methods, the abstract focus mainly on cell-based methods. I suggest revising the abstract to reflect the scope of the review.

- The abstract has been completed accorgingly.

Throughout the manuscript: Please define all abbreviations the first time they are used.

- Abbreviations have been completed.

Page 2, line 70: Please replace “Diagnostic of” with “Diagnosis of”

- Done

Page 2, lines 80-83: Please mention that treatment also includes supportive care, intubation, and mechanical ventilation when necessary.

- Done

Page 2, lines 86-87: The sentence “In order to replace animal use…” is not clear. Maybe it is missing a few words? Please clarify.

- Done

Page 2, lines 90-92: The sentence repeats “BoNT detection in clinical specimens and foods”; please reword.

- Done

Page 2, line 84: Suggest changing the title of this section to “laboratory confirmation of botulism” and combining it with the previous section “Diagnosis of botulism”. Both sections refer to the same.

- Done

Page 3, lines 97-104: This section does not seem to fit here, as the review is about detection of BoNT, and this section refers to DNA based methods to identify C. botulinum.

- This section has been reduced.

Page 3, lines 130-138: the manuscript mentions the requirements of the European Pharmacopea for potency determination. What are the requirements outside the EU? How do pharmaceutical companies determine potency of BoNT in the US, for example?

This has been completed.

Page 7, lines 262-20: Have these methods been used to test clinical specimens and/or foods? Please clarify.

- Yes but very limited data (avian blood, milk and different beverages : Cohen 2016)

Page 8, lines 288-297: Have these methods been used to test clinical specimens and/or foods? Please clarify.

- Yes but limited data to date.

Page 9, lines 374-384: Has this method been used to test clinical specimens and/or foods? Please clarify.

- No data found to date.

Page 10, lines 386-390: Has this method been used to test clinical specimens and/or foods? Please clarify.

- No data found to date.

Page 10, line 399: Please reword the title of this section.

- Done

Page 12, lines 409-436: Although the manuscript covers several test methods, the Discussion section focus mainly on cell-based methods. I suggest revising this section to reflect the more general scope of the review.

- Done
